# Toxicity Tolerance in the Carcinogenesis of Environmental Cadmium

**DOI:** 10.3390/ijms25031851

**Published:** 2024-02-03

**Authors:** Aleksandar Cirovic, Soisungwan Satarug

**Affiliations:** 1Institute of Anatomy, Faculty of Medicine, University of Belgrade, 11000 Belgrade, Serbia; aleksandar.cirovic@med.bg.ac.rs; 2Kidney Disease Research Collaborative, Translational Research Institute, Woolloongabba, Brisbane, QLD 4102, Australia

**Keywords:** cadmium, cancer, cell transformation, copper, intestinal absorption, iron store status, zinc transporters

## Abstract

Cadmium (Cd) is an environmental toxicant of worldwide public health significance. Diet is the main non-workplace Cd exposure source other than passive and active smoking. The intestinal absorption of Cd involves transporters for essential metals, notably iron and zinc. These transporters determine the Cd body burden because only a minuscule amount of Cd can be excreted each day. The International Agency for Research on Cancer listed Cd as a human lung carcinogen, but the current evidence suggests that the effects of Cd on cancer risk extend beyond the lung. A two-year bioassay demonstrated that Cd caused neoplasms in multiple tissues of mice. Also, several non-tumorigenic human cells transformed to malignant cells when they were exposed to a sublethal dose of Cd for a prolonged time. Cd does not directly damage DNA, but it influences gene expression through interactions with essential metals and various proteins. The present review highlights the epidemiological studies that connect an enhanced risk of various neoplastic diseases to chronic exposure to environmental Cd. Special emphasis is given to the impact of body iron stores on the absorption of Cd, and its implications for breast cancer prevention in highly susceptible groups of women. Resistance to cell death and other cancer phenotypes acquired during Cd-induced cancer cell transformation, under in vitro conditions, are briefly discussed. The potential role for the ZnT1 efflux transporter in the cellular acquisition of tolerance to Cd cytotoxicity is highlighted.

## 1. Introduction

Cadmium (Cd) is a metal with no nutritional or physiological value in humans, but it is found in most people because it is a contaminant in nearly all food types; as such, dietary exposure is unavoidable for most people [1,2,3,4,5]. Active and passive smoking are additional non-workplace exposures to Cd [6]. Airborne Cd is increasingly detectable, especially in urban areas of many countries [7]. Consequently, an inhalation exposure and the existence of a nose-to-brain entry route cause further concern [8,9].

Cd exists primarily in a divalent state (Cd^2+^) and as a redox inert metal; it does not undergo valency change [10,11]. However, compared to an essential metal zinc (Zn^2+^), Cd has a higher binding affinity for sulfur ligands that include the thiol (-SH) functional group of glutathione, zinc-finger transcription factors, and many other proteins [10,11,12,13]. It is well documented that Cd interacts with the cysteine thiols of metallothionine (MT), a group of low-molecular-weight proteins, capable of sequestering zinc, copper, and Cd [11,12,13]. Also, Cd induces mitochondrial dysfunction, but it does not directly damage DNA [14,15]. Likely, the manifestation of carcinogenic effects of Cd are through its interactions with essential metals and proteins, i.e., DNA repair enzymes [14,15,16]. Cd is viewed as a modulator of human gene transcription [15].

Chronic exposure to Cd produces a broad range of adverse health effects. This is evident from epidemiological studies that link an incremental risk of infertility, cancer, diabetes, chronic kidney disease, osteoporosis, and non-alcoholic fatty liver disease to Cd exposure in the general populations of many countries [1]. The overall health impact of Cd comes from cohort studies which implicate Cd exposure in an increased mortality from all causes [17,18,19,20], heart failure [21,22], and cancer [23,24]. 

An elevated Cd body burden, smoking, low-level physical activity, and low serum antioxidant nutrient lycopene were found to be associated with an increase in deaths from all causes in the United States [16]. These modifiable mortality risk factors were identified from an analysis of 249 health indicators, included in the U.S. general population studies, known as the National Health and Nutrition Examination Survey (NHANES) [17]. 

Notably, Cd was found, through mediation analysis, to be responsible for most of the adverse health effects of active and passive smoking [25]. This result is predictable because of the long residence time of Cd in cells in the absence of a normal excretory mechanism [1]. The Cd concentration in cells increases with age (duration of exposure). The estimated half-life of Cd in the body varies between 7.4 and 30 years; the lower the body burden, the longer the half-life of Cd [26,27,28,29]. In comparison, nicotine, another constituent of cigarette smoke, can be metabolized rapidly, and is eliminated completely through urinary excretion [30,31,32].

The present review was to provide an update of the knowledge on metal transporters involved in the assimilation of dietary Cd and the key determinants of the intestinal absorption rate, namely body iron store status, iron deficiency, and diet quality. Epidemiological data connecting an enhanced risk of breast and liver carcinomas to Cd exposure are provided. Target iron supplementation is discussed as a strategy to reduce Cd accumulation and a consequential reduction in breast cancer risk among a highly susceptible group of women. Special emphasis is on Cd-induced malignant cell transformation, and the potential mechanism(s) underlying resistance to cell death and other cytotoxicity due to Cd, which are acquired during cell transformation. 

## 2. Dietary Sources and Determinants of the Body Burden of Cadmium

### 2.1. Estimation of Exposure to Cadmium in the Human Diet

Average and high dietary Cd exposure levels of 30 and 93.5 µg/day were estimated from a typical Australian diet, and the levels of Cd in foods (Table 1).

As the data in Table 1 indicate, cereals and vegetables contributed mostly to total dietary Cd exposure. These sources of dietary Cd were also identified when the Australian market basket survey data were used [2]. Of total dietary Cd exposure ranging between 9 and 15 µg/day, potatoes, wheat, cocoa, and meat contributed 46, 16, 12, and 7% of total dietary Cd exposure, respectively, while crustaceans, liver, peanuts, and vegetables each contributed 2–3%, adding a further 11% to total dietary Cd exposure [33].

The average dietary Cd exposure level in Sweden was 10.6 μg/day with 40–50% of Cd coming from staple foods (potatoes and wheat), while a high dietary Cd exposure level was 23 μg/day, with the additional Cd coming from seafood and spinach [3]. The average dietary Cd exposure in France was 11.2 µg/day with 35% of Cd coming from bread products and another 26% from potato-based products, while a high Cd exposure level was 18.9 µg/day with the additional Cd coming from mollusks and crustaceans [4].

### 2.2. The Intestinal Absorption of Cadmium

Since metals cannot be synthesized nor destroyed by cells, all physiologically required metals must be sourced from the external environment. However, the environmental levels of most metals are generally low, termed micronutrients; consequently, highly specific and efficient uptake mechanisms and transport pathways are necessary to ensure an optimal supply of metals in accordance with physiological demand [34,35,36,37,38]. Also, to prevent the harmful effects of deficiency or over supply (overload), the body content of metals, especially iron, is tightly regulated. Since there is no biochemically active mechanism to eliminate metals, the regulation of the entry of metals is a dominant strategy to maintain metal homeostasis. 

Likely, the transporters and ion channels involved in the absorption of Cd by enterocytes include those for essential metals; iron, zinc, calcium, manganese, copper, and cobalt (Fe, Zn, Ca, Mn, Cu, and Co) detailed below. Consequently, the intestinal absorption of Cd and its transport follow closely those of essential metals. 

Studies from Japan suggested that the absorption rates of Cd among women were 24–45% [39,40], but lower Cd absorption rates of 3–7% were assumed in a conventional health risk assessment of Cd [41]. The assumed low intestinal absorption rates resulted in miscalculation and erroneous conclusions concerning the health risk posed by dietary Cd exposure. As detailed below, Cd can be taken up by several metal transporters, and in addition, Cd complexed with MT and phytochelatin can be assimilated through transcytosis and receptor-mediated endocytosis [42,43,44]. 

### 2.3. Metal Transporters Involved in Cadmium Absorption

Collective human and experimental data suggest that Cd assimilation involves the divalent metal transporter 1 (DMT1), certain calcium channels (TRPV6), and metal transporters of the Zrt-/Irt-like protein (ZIP) family, notably ZIP14 [45,46,47,48,49]. ZIP14 was also found to mediate iron absorption [48,49] and possibly the exit of Cd from the enterocyte [49]. The zinc–iron interactions have been observed in studies of Bangladeshi women [50,51]. Genome-wide association and genetic linkage studies are consistent with the roles of zinc and iron transporters and proteins of iron homeostasis as determinants of blood and urinary Cd levels [52,53,54].

Expression levels of metal transporters in the intestine are highly variable, depending on age, nutritional status, and physiological requirements for the metals, e.g., in pregnancy or lactation [50,51,55,56]. 

Normally, the body burden of an individual contaminant is a balance between the absorption and elimination rates. Studies in the 1970s estimated that 0.001–0.005% of the total amount of Cd in the body was excreted each day [23,24]. An extremely slow excretion rate of Cd implies that Cd is a cumulative toxicant, in which its level increases with age (duration of exposure) and that the intestinal absorption rate of Cd essentially determines the body burden. In the absence of a therapeutically effective chelation for Cd, the most practical and effective preventive measure is to minimize the absorption rate of Cd. 

Although our knowledge on Cd absorption in humans remains fragmentary, the most likely transporter proteins/carriers involved are depicted in Figure 1.

### 2.4. Determinants of Cadmium Absorption

#### 2.4.1. Body Iron Stores and Iron Deficiency

The absorption rate of iron is increased when the body iron stores are low or depleted, indicated by serum ferritin concentrations less than 20 and 30 µg/L, respectively [56,57,58]. The absorption rate of iron is increased markedly in the iron deficient state, indicated by hemoglobin concentrations and body iron indicators, namely serum ferritin, free erythrocyte protoporphyrin, and % transferrin saturation [58]. 

Universally, an inadequate iron store status and iron deficiency are more prevalent in females than males, and are more common in children than adults, as described below. 

The prevalence of iron deficiency amongst Australian women, aged 25–49 years, was 20.3% [59]. The percentages of iron deficiency among Korean adolescents and women aged 19–49 years were 36.5% and 32.7%, respectively [60].

The prevalence of iron deficiency among U.S. children aged 3–19 years enrolled in NHANES 1999–2002 (*n* = 5224) was 7% [58]. An association between blood Cd and iron status was found to be strongest in females, 16–19 years of age, and high blood Cd levels (≥0.5 μg/L) were 1.74 times more prevalent among those with iron deficiency [57]. Lower serum ferritin concentrations were associated with higher blood Cd and manganese among adolescents (12–17 years) in NHANES 2017–2018 [58].

Non-smoking Norwegian women with low body iron stores had higher blood Cd, Mn, and Co than similarly aged women whose body iron stores were within a normal range [61]. One in four women with iron deficiency had elevated blood levels of Cd and manganese. In a study of young women, Finley (1999) noted an increased manganese absorption rate in those with low serum ferritin concentrations, which indicates low iron stores [62].

In a study from Thailand, women (mean age 30.5) with low body iron stores were found to excrete Cd at a level 3.4-fold higher than similarly aged women who had adequate iron stores [63]. In a study from Korea, the mean for blood Cd levels in women with iron deficiency was 1.53 µg/L, which was 48.5% higher than those with normal iron status [60].

#### 2.4.2. Dietary Factors: Diet Quality

The intestinal absorption of Cd has rarely been studied. The findings from limited literature reports are summarized herein. Cd in a shellfish diet was shown to be bioavailable in the study by Vahter et al. (1996) who found Cd intake to be 11 μg/day for women who consumed a mixed diet and 28 μg/day for those who had a high shellfish diet [64]. No differences in blood or urine Cd levels were observed between the two groups. However, there were 63% and 24% increases in blood and urine Cd among those consuming the high shellfish diet who had depleted iron stores (plasma ferritin levels < 20 μg/L), compared with those who consumed mixed diets and had the same low body iron stores. Thus, these studies strongly suggest that Cd in shellfish is bioavailable and that long-term consumption does result in a higher body burden of Cd. 

A study from the Torres Strait (Australia) reported a 1.4-fold increment of urinary Cd was associated with the consumption of turtle liver and kidney and locally gathered clams, peanuts, and coconuts [65]. The sum of these foods, heavy smoking, age, and waist circumference accounted for 40% of the Cd body burden.

A study from British Columbia observed that elevated urinary Cd excretion was associated with a duration of oyster farming of at least 12 years plus an average consumption of 18 oysters/week (87 g/week) [66]. The estimated Cd exposure from the consumption of oysters was 174 μg/week (24.8 μg/d), and the mean urinary Cd was 0.76 μg/g creatinine (range: 0.16–4.04 μg/g creatinine). This mean urinary Cd was approximately twofold higher than the mean urinary Cd of 0.35 μg/g creatinine, recorded for non-smokers, aged 20–75 years, in the Canada Health Measure Survey [67].

In experimental studies, the levels of calcium, zinc, iron, and fiber in the diet can affect the absorption and accumulation of Cd [68,69,70]. Levels of Cd accumulation in the liver and kidneys were reduced by 70–80% after 8 weeks of feeding with Cd-tainted food supplemented with calcium, phosphorus, iron, and zinc [70]. The reduction of Cd accumulation was attributed to the presence of iron (Fe^2+^). The addition of vitamin C improved iron uptake but did not reduce Cd accumulation. Iron in combination with calcium, phosphorus, and zinc produced the most remarkable effect. 

In an experimental study in mice, a diet deficient in iron or calcium led to much greater Cd accumulation in the kidneys than a diet deficient in copper, zinc, or manganese [71,72]. Renal Cd accumulation levels strongly correlated with the intestinal expression of calcium transporter1 (CaT1) and metallothionein isoform1 (MT1). Younger mice accumulated more Cd in their kidneys than older mice, when fed with a calcium-deficient diet. In addition, an iron-deficient diet caused a greater increase in hepatic Cd accumulation than any other diet tested.

In experimental animals, a diet high in phytate (fiber) enhanced the absorption and renal retention of Cd [69]. Such effects require further study because they could endanger humans with high fiber intake. Supplementation of food commodities with citric acid, which is a common practice, enhances the bioavailability of zinc, and may also result in increased Cd absorption [73].

Flavonoids and curcumin in the diet may reduce Cd absorption through the formation of complexes with Cd [74,75]. A study in mice showed that Cd administration with curcumin resulted in lower Cd levels in the blood and organs (liver, kidney) than in mice that received Cd only [74,75]. Regular consumption of curcumin was linked to lowered blood Cd and lowered risk of hypertension in a Korean study [76].

#### 2.4.3. Genetic Factors: The ZIP8, ZIP14, TfR, and H63D Variants

Argentine Andean women carrying the GT or TT genotype of the rs4872479 ZIP14 variant had 1.25-fold higher blood Cd levels than GG genotype carriers. Also, the Andean women carrying the AG or GG genotype of the rs10014145 ZIP8 variant had 1.18-fold higher blood Cd levels than those carrying the CC genotype [52]. Transferrin receptor variant (TFRC rs3804141) was associated with elevated urinary Cd [53].

In patients with hemochromatosis or iron deficiency, there was an increase in the intestinal expression of the iron transporter DMT1 which creates a greater capacity for the absorption of iron [34,38]. Because DMT1 has a higher affinity for Cd^2+^ than Fe^2+^ [77], an increased expression of the DMT1 in people with hemochromatosis or iron deficiency, in general, would provide them with a greater capacity to absorb iron and possibly Cd.

A longitudinal cohort of U.S. men, aged 51−97 years, known as the Normative Aging Study, linked high toenail Cd levels to the homozygous H63D variant alleles of the hemochromatosis (HFE) gene and to low hemoglobin levels [78]. 

### 2.5. Summary: Determinants of Cadmium Absorption Rate

Body iron store status is a major determinant of Cd absorption rate, and thus Cd body burden. Indicators of low body iron stores are associated with elevations of urinary and blood Cd levels in children, adolescent females, and women of reproductive age. Body iron stores in these populations are universally lower than in male counterparts. Habitual consumption of high-Cd-containing foods is an important determinant of the body burden of Cd. A systematic evaluation of the effects on micronutrient status on Cd body burden showed strong evidence for inverse relationships between zinc and iron status and Cd body burden [79].

## 3. Cadmium as a Multi-Tissue Carcinogen: Epidemiologic Studies

The International Agency for Research on Cancer (IARC) listed Cd as a human lung carcinogen in 1993, based on the incidence of lung cancer in workplace exposure settings [80]. This IARC assertion gained little attention in the evaluation of health risk of dietary Cd exposure because of the perception that the cancer risk of Cd was applied only to an inhalational exposure to relatively high doses of volatile metallic and oxide forms of Cd fumes and dusts, experienced by workers. 

This section summarizes the epidemiological data suggesting that Cd may induce the formation of carcinomas in many organs, not limited to the lung or exposure routes. 

### 3.1. Cadmium and Human Cancer: Evidence from a Meta-Analysis

A summary of the carcinogenic risk of Cd evaluated through a meta-analysis is found in Table 2.

### 3.2. Impact of Cadmium on Cancer Risk

The carcinogenic risk assessment of Cd provides evidence that lifelong Cd exposure may increase the risk of neoplasm formation in the lung [81,82], kidney [83], and pancreas [84]. 

Cd exposure was found to be a particularly strong risk factor for breast cancer, only when the urinary excretion of Cd was used as a measure of cumulative long-term exposure (body burden) [85,86]. In contrast, dietary Cd exposure was not associated with breast cancer risk in postmenopausal women [88]. A similar non-association between dietary Cd exposure and breast cancer risk was observed in another two analyses [89,90]. This non-association of dietary Cd exposure and breast cancer risk is not unexpected; as we have elaborated in Section 2, the amount of Cd that reaches a target depends largely on absorption rate, not just the amount of Cd in the diet. Thus, the utility of dietary exposure, as an indicator of exposure, in health risk assessments, is questionable.

The central dogma states that for a toxicant to exert its effects, it must first enter the body, and reach its target cells. Andjelkovic et al. (2012) analyzed the levels of Cd in 55 breast tissue samples, obtained during surgery, where they found almost four times higher Cd quantities in cancer tissues, compared to the surrounding non-cancerous tissues [91]. In addition, scalp hair and toenail samples from women with breast cancer contained higher Cd and nickel than non-cancer controls [87].

### 3.3. Cadmium and Hepatocellular Carcinoma

The carcinogenic potential of Cd accumulation in the liver has attracted little attention although hepatocellular carcinoma (HCC) is the second commonest cause of cancer death worldwide [92,93,94]. The causes of HCC are chronic infection with hepatitis B virus (HBV) or hepatitis C virus (HCV), and dietary exposure to aflatoxin B1 (AFB1), a hepatocarcinogen produced by *Aspergillus flavus*, grown on peanuts, corns, and soy products [92,93,94]. Mutation in the tumor suppressor protein 53 gene, caused by the G:C-to-T:A transversion, is the molecular signature of HCC in areas with high rates of HBV infection plus AFB1 exposure [95]. 

Other HCC causative conditions are alcoholic steatohepatitis, non-alcoholic steatohepatitis (NASH), iron overload, excessive copper accumulation (Wilson’s disease), congenital α1-antitripsin deficiency, autoimmune hepatitis, and acquired and congenital glycogen storage diseases [96,97,98]. Studies from the United States and Korea, described below, observed associations of increased risk of NASH and Cd exposure, thereby suggesting that Cd may promote NASH development, which may progress to liver fibrosis and HCC.

HCC risk factors are male gender, old age, long-term use of oral contraceptives, and the immunosuppressant azathioprine [99,100,101]. A genetic defect in the enzyme glucose-6-phosphatase, essential for glucose release from liver glycogen, is another known risk factor in HCC and hepatic adenomas [102]. 

In 1990, Campbell et al. first observed an association of liver cancer mortality with exposure to Cd in foods of plant origin, but not AFB1 [103]. The reason as to why plant food Cd may have high carcinogenic potential is not known. Probably, it is due to an amount of Cd that forms a complex with plant phytochelatin, and as discussed earlier that CdPC can be absorbed intact through transcytosis [42] and receptor-mediated endocytosis [43,44]. Campbell et al. also observed an association of HCC mortality with the prevalence of hepatitis B virus surface antigen (HBsAg+) carriers, plasma cholesterol, and alcohol consumption. 

Recently, a connection between liver cancer and Cd exposure gained some interest, evident from two studies from China [104,105], and a report of a cohort study from Japan [20], which observed increased deaths from carcinoma of the liver and pancreas in men, who showed signs of kidney tubular cell damage and impaired tubular reabsorption function [20]. Previous analysis of the data from the same subjects revealed a dose–response relationship between the excretion levels of Cd and those signs of nephrotoxicity. 

In Australian autopsy studies, increased hepatic sequestration levels of zinc and copper were found in human liver samples containing Cd ≥ 1 µg/g wet liver weight [106,107,108]. Hepatic levels of copper rose 45–50% in persons with high Cd body burden, compared to similarly aged subjects who had a medium Cd body burden [107]. Excessive liver copper and iron accumulation are known HCC causative factors [92,97].

Evidence for the hepatotoxicity of environmental Cd is summarized in Table 3.

Increased risks of liver inflammation in both U.S. men and women were observed with urinary Cd levels ≥ 0.6 µg/g creatinine as were the risks of non-alcoholic fatty liver disease (NAFLD) and non-alcoholic steatohepatitis (NASH) in men [109]. Signs of the hepatotoxicity of Cd were noted in U.S. adults [110]. The hepatotoxicity of Cd was more pronounced in boys than girls [111].

Similarly, evidence for the effects of Cd on the liver were observed in Korean population studies [112,116]. Increased risks of liver fibrosis, NAFLD, and NASH in the representative Korean general population were also found to be associated with blood and/or urinary Cd levels [114,115]. 

### 3.4. Cadmium and Other Types of Cancer 

Other types of cancer found to be associated with Cd exposure were endometrial cancer [117], acute myeloid leukemia [118], urinary bladder cancer [119], oral and gastrointestinal carcinomas [120,121], and nasopharyngeal and pharyngeal cancers [122]. 

There was little evidence for the carcinogenicity of Cd in the prostate [123] although plasma and prostate tissue samples from those with prostate cancer contained Cd levels higher than those from controls [124]. 

### 3.5. Implications for Heath Risk Calculations 

Cancer risk evaluations through systematic reviews and meta-analyses, shown in Table 2, have provided evidence that Cd may be involved in the development of carcinomas in many organs. The enhanced cancer risk was observed at a low Cd body burden, indicated by urinary Cd excretion below 1 μg/g creatinine. For example, the risk of breast cancer rose 66% for each 0.5-μg/g creatinine increment of urinary Cd excretion [85].

In a case–control study, a dose–response relationship between breast cancer risk and Cd exposure was apparent, when individuals with urinary Cd excretion levels ≤ 0.26 were compared with those with ≥0.58 μg/g creatinine, suggesting a 2.29-fold increase in breast cancer risk [125]. In another cohort study of U.S. women, Cd excretion levels ≥ 0.37 μg/g creatinine were associated with a 2.5-fold increase in the risk of breast cancer [126].

These data cast considerable doubt on the current practice of the health risk assessment of Cd, which relies solely on kidney tubular dysfunction, defined as urinary excretion of β_2_-microgloblin (β_2_M) ≥ 300 μg/g creatinine, as the adverse health outcome of concern [41]. Based on this β_2_M endpoint, the upper limits of tolerable intake and acceptable excretion of Cd are 0.83 µg/kg body weight/day and 5.24 µg/g creatinine, respectively. A suggested tolerable intake level is equivalent to 58 µg/day for a 70 kg person. Apparently, these exposure guidelines and the nephrotoxicity threshold levels of Cd are not sufficiently low to afford health protection. In theory, the toxicity threshold level should be based on the most sensitive endpoint with consideration given to susceptible subpopulations [127]. 

### 3.6. Cadmium, Breast Cancer, and Iron Supplements 

As an estimate for 2021, there were 281,550 new cases of breast cancer in the U.S. which formed 30% of all cancer diagnoses in women [128]. Thus, the proportion of breast cancer is approaching epidemic levels.

Risk factors identified for breast cancer are female gender, advanced age, positive family history of breast cancer, commencing periods at a younger age, no history of pregnancy, and menopause onset at an older age [128]. Obesity, low physical activity, and alcohol abuse were additional risk factors identified, which were modifiable. 

It is noteworthy that many cases of female breast cancer do not have any of these identified risk factors. Therefore, there is a need to identify all other factors/conditions that may promote the development of breast carcinoma, from which prevention strategies and targeted therapies could be formulated.

In a study from South Korea, including 454,304 adults of which 41,947 (9.2%) were anemic, an increased risk of breast cancer was noted among those with anemia [129]. In another Korean study, female residents of rural areas were found to have a lower risk of breast cancer, compared to women who resided in urban communities [130]. It was suggested that urban residents may have had exposure to higher Cd levels in the dust than rural residents [7]. Collectively, the data from these two Korean studies could be interpreted to suggest that an increased risk of breast cancer among anemic women who lived in urban communities was due an increased exposure to Cd in the dust.

A causal relationship between pre-existing anemia and living in an urban area with consequential breast cancer development remains uninvestigated, but it is fundamentally important because almost a quarter of the world’s population has been affected with anemia, whereas women in their reproductive period are broadly anemic. Also, the majority of the world’s population lives in cities. The evaluation of cancer risk through meta-analyses has identified an increased body burden of Cd as a strong risk factor for breast cancer [85,86].

Based on the literature reports of high blood and urinary Cd (body burden) among those with low body iron stores and iron deficiency (Section 2.3), we speculate that environmental exposure to Cd through diet, smoking, and polluted air could contribute to the development or progression of breast cancer, especially in at-risk subpopulations. Arguably, supplementing iron to women during the reproductive period, particularly those with co-existing anemia, would reduce the body Cd load, and the risk of breast cancer.

### 3.7. Role of Cadmium in the Genesis of Breast Cancer 

Understanding the precise pathophysiological mechanisms could be useful to predict cancer risk and formulate rational target therapies. We briefly discuss the potential role of Cd in the genesis of breast cancer. Since Cd does not directly damage DNA [14,15], it is suggested that Cd causes breast cancer by several mechanisms, notably by dampening DNA repair enzymes [14,15,16]. Cd can mimic estrogen activity and is viewed as a xenoestrogen [131]. Siewit et al. showed that Cd stimulates estrogen receptor ɑ (ERɑ) and promotes glandular cell growth [132].

In the case when HIF-1 is overexpressed in cancer tissues, the cancer is more invasive and patients face poor prognosis [133]; additionally, if cancer cells highly express TfR1, patients face unsatisfactory outcomes [134,135]. For both cases, the mechanisms are the same, namely, that TfR1 is a HIF-inducible gene and is highly expressed when HIF is generated within the cell. TfR1 mediates the absorption of all the metals that were bound to transferrin which can bind both iron and Cd. 

The accumulation of Cd in malignant cells may deepen cancer hypoxia, which is linked with worse outcomes, generates additional mutations, and promotes metastasis. Cd is pro-carcinogenic once it reaches breast tissue, and individuals with anemia are heavily loaded with Cd. 

Abul-Husn et al. found that 1/139 individuals have a pathogenic variant of the BRCA1/2 genes. Thus, there are many individuals who have an aberrant form of this gene [136]. Mutations of the BRCA1 and BRCA2 genes lead to an increased risk of breast cancer. Individuals who are heterozygotes for BRCA1 or BRCA2 are probably born with this mutation, and are more susceptible to developing cancer, but must acquire a second mutation during their lifespan. Other environmental factors may be responsible for the generation of the second mutation; for example, if an individual with one BRCA1 mutation was exposed to radiation, then the radiation could generate the second mutation and breast cancer could occur.

Inhibitors of HIF-1 show positive results in breast cancer treatment [137]. In addition to its potential role as an initiator of cancer cells, Cd may promote the proliferation, migration, and invasion of breast cancer cells [138]. Accordingly, we speculate that if individuals with breast cancer were supplemented with iron, then malignant cells will uptake more iron and less circulating Cd, given that both metals are bound to transferrin and are taken up by TfR1. These scenarios will result in a reduced accumulation of Cd in cancer tissue, diminished cancer invasion, and an improved outcome.

By supplementing iron in women during the reproductive period, particularly those with co-existing anemia, we could reduce the body Cd load, reducing the risk of breast cancer. Also, individuals with breast cancer must immediately stop smoking and we recommend that regular breast treatments (e.g., surgery, HIF-1 inhibitors) should be accompanied by iron supplementation. The beneficial effects of iron should be confirmed in controlled trials. 

The mitotic crossover could generate a second mutation as well, as Ivanovski et al. discussed, but if the mechanisms of DNA repair function properly, then the chances for the second mutation are negligible. Cd is an inhibitor of DNA repair and consequently, minor errors caused by mitotic crossover or by DNA replication which could affect any part (gene) of the chromosome could lead to the occurrence of a second mutation and the consequent loss of heterozygosity. Although DNA replication is a highly accurate process, mistakes may occur leading to known point mutations. If the error affects the opposite BRCA1/2 gene (since mutation number one already exists) both genes will encode a non-functional protein. As we already discussed, the final step in this chain of events is lacking in DNA repair, and cadmium can make it function improperly [139,140]. Cd is toxicant, and its impacts are dose-dependent, so individuals who are heavily loaded with Cd are at a pronounced risk that their DNA repair mechanism will not identify the error caused by DNA replication and a second mutation might be generated.

We hypothesized that the accumulation in breast tissue contributes to the occurrence of the second mutation in the BRCA1 or BRCA2 gene as depicted in Figure 2.

## 4. Cadmium as a Multi-Tissue Carcinogen: Experimental Studies

This section summarizes the results from a standard 2-year bioassay, which was designed to evaluate the tumorigenesis of a suspected carcinogen over the lifespan of rats and mice. However, the high cost involved in conducting the 2-year bioassay has led to the idea of generating cancer cells from non-tumorigenic human cells. Accordingly, this section summarizes a successful attempt to produce cancer cells under in vitro conditions. The molecular basis for the tolerance to Cd toxicity and metabolic phenotypes acquired during cancer cell transformation are highlighted. 

### 4.1. Cadmium and Tumor Formation in Mice 

In a 2-year bioassay that mimics a lifetime exposure scenario of the rodent and murine species, Waalkes and Rehm (1994) found that the tumorigenic actions of Cd depended on the genetic background or the strain of animals used [141,142,143,144]. 

In their landmark experiment [141], Cd was administered to the DBA/2NCr (DBA) and NFS/NCr (NFS) mice at 8 weeks of age (an adult age) by subcutaneous injection, as a single dose of 40 µmol/kg or as a weekly dose of 40 µmol/kg for 16 weeks (16 × 40 µmol/kg). Thereafter, the mice were observed for two years. 

HCC and hepatic adenomas were not found in any of the DBA mice, but were found in 1/15 control NFS mice, and 9/27 NFS mice, treated with weekly repeated Cd administrations. Sarcoma at the injection site occurred in 9/35 NFS mice with repeated exposure to Cd. Distinctively, the incidence rates of testicular tumors in the two strains were similar. Intriguingly, lung tumors were found in the NFS mice treated with a single dose of Cd.

The genetic basis contributing to different sensitivities or resistance to the tumorigenic effects of Cd among murine strains has not been fully investigated. However, the high testicular toxicity of Cd has been linked to ZIP8, a zinc transporter, responsible for the uptake of zinc, manganese, and Cd by cells [46,47,48]. These findings accentuate the influences of metal transporters in determining the intracellular concentration of Cd, and thus its toxicity manifestation.

### 4.2. The Genesis of Lung and Liver Carcinomas after Cadmium Exposure 

A single-dose subcutaneous injection of Cd at 40 µmol/kg was sufficient to cause tumor formation in the lungs of the NFS mice, while HCC and hepatic adenomas were formed in the same strain, NFS mice, with weekly repeated Cd administrations [141].

In comparison, the induction of lung tumors by benzo(*a*)pyrene, a ubiquitous environmental carcinogen present in cigarette smoke and charred meat products, would require a repeated exposure regime due to the rapid rates of metabolism and elimination [145]. Furthermore, benzo(*a*)pyrene becomes a potent DNA-damaging agent only after conversion to benzo(*a*)pyrene-7,8-diol-9,10-epoxide (BPDE) by the action of microsomal cytochrome P450 (CYP) enzymes, CYP1A1/1A2 and CYP1B1 [145]. Repeated exposure results in the accumulation of BPDE–DNA adducts to a critical level causing mutations that affect the expression of oncogenes and tumor suppressor genes [145]. 

This is analogous to the G:C-to-T:A transversion mutation caused by AFB1–DNA adducts, leading to the inactivation of tumor suppressor protein 53 gene [95]. This mutation represents the molecular signature of HCC in areas with high dietary exposure to AFB1 and a relatively high rate of HBV infection [95].

### 4.3. Induced Formation of Cancer Cells

There has been a long history of attempting to generate cancer cells in vitro from immortalized human cell lines that do not form tumors in nude mice, so-called non-tumorigenic or non-neoplastic cells. If successful, this in vitro cell transformation could be an alternative to a high-cost 2-year bioassay. Such an attempt proved to be difficult for carcinogens like benzo(*a*)pyrene and AFB1. As discussed above, the tumorigenic actions of benzo(*a*)pyrene and AFB1 depend on metabolic activation, involving specific CYP enzymes. 

Distinctively, the carcinogenic action of Cd does not require any CYP enzymes, and cancer cells have now successfully been produced in vitro. The human and rodent cell lines, Cd concentration, and exposure duration used in Cd-induced cell transformation experiments, and the histologic phenotypes of cancer cells and tumors generated in nude mice are summarized in Table 4. 

The immortalized, non-tumorigenic human cells susceptible to the carcinogenicity of Cd include UROtsa urothelial cells [146,147], MCF-10A breast epithelial cells [148], BEAS-2B bronchial epithelial cells [149], HPL-1D peripheral lung epithelium [150], and HPDE pancreatic ductal epithelial cells [151]. The cell line of animal origin, found to be susceptible to Cd carcinogenesis, is the rat liver epithelial TRL1215 cell line [152,153].

The generation of cancer cells simply by exposure of various cell types to low-level Cd over a long period of time (Table 4) provides further evidence that Cd is a muti-tissue carcinogen as it has been observed in both Cd-exposed humans (Table 2) and Cd-exposed mice (Section 4.2). The Cd-transformed cells (transformants) have been of utility to investigate the molecular fingerprints that can be linked to cellular adaptive responses, enabling the transformant to resist cell death, and acquired cancer cell phenotypes such as metabolic reprogramming to glycolysis dominance, known as the Warburg effect [154,155,156]. The notable impact of Cd on cellular gene expression profiles has been studied extensively using various types of cancer cells, including MCF-7 breast cancer cells, A549 lung cancer cells, and HepG2 hepatocellular carcinoma cells [157,158,159].

### 4.4. Acquired Resistance to Cadmium-Induced Cell Death: Role of ZnT1 

Among the susceptible cells, the UROtsa urothelial cell line is noteworthy because of the limited human cell models of human urinary bladder cancer, which has a high recurrence rate. The UROtsa cell line is a non-neoplastic cell line, which shows phenotypic and morphologic characteristics resembling primary transitional epithelial cells [160]. This cell line was derived from the epithelium of the ureter of a 12-year-old female donor, immortalized with SV40 large T-antigen [161]. Chronic exposure to Cd caused the UROtsa cells to undergo neoplastic transformation, expressing the phenotype characteristic of transitional cell carcinoma of the bladder [146].

Resistance to cell death is one of the common cancer cell phenotypes. In cell transformation experiments (Table 4), the Cd concentrations used range from 1 to 5 µM. In comparison, in the investigation of the effects of Cd on the genes expressed by various cancer cells, typically the Cd concentrations used were between 10 and 100 µM [157,158,159]. Thus, cancer cells appeared to tolerate higher Cd concentrations, compared to non-tumorigenic immortalized cells. 

A reduced Cd accumulation has been suggested to be the reason for resistance to cell death. In an experiment using TRL1215 rat liver epithelial cells, the pretreatment of cells with cyproterone, a synthetic steroidal antiandrogen with a structure related to progesterone, decreased sensitivity to Cd through a decreased accumulation of Cd [162]. However, the molecular basis for a decrease in Cd accumulation was not investigated. It was shown in another study that silencing the expression of ZnT1 results in an increased Cd accumulation, and thus enhanced Cd toxicity [163]. ZnT1 is an efflux transporter that mediates the extrusion of both zinc and Cd, thereby lowering intracellular Cd concentrations [164].

In a more recent study, the expression of ZnT1 in UROtsa cells was increased by 1 µM Cd, which was the same concentration that was used in the transformation experiment [146]. Pretreatment of UROtsa cells with an inhibitor of glutathione biosynthesis (buthionine sulfoximine) diminished ZnT1 induction with a resultant increase in sensitivity to the cytotoxicity of Cd [165]. Conversely, pretreatment of UROtsa cells with an inhibitor of DNA methylation, 5-aza-2′-deoxycytidine (aza-dC), did not change the extent of ZnT1 expression levels in Cd-treated cells. The induced expression of ZnT1 that remained impervious in cells treated with aza-dC coincided with resistance to Cd cytotoxicity [165]. 

The expression of ZnT1 in A549 human lung cancer cells rose after exposure to 20 µM Cd [158]. The expression of ZnT1 in Cd-transformed HPL-1D human peripheral lung epithelium also rose together with ZnT5 and ZIP8, leading to a reduced Cd accumulation [150]. Collectively, these experimental data underscore the significance of metal transporters, in particular, the ZnT1 efflux transporter, in the acquisition of tolerance to Cd by various cancer cells [166]. 

## 5. Conclusions

Epidemiologic and experimental data provide compelling evidence that Cd is a multi-tissue carcinogen. Systematic reviews and meta-analyses indicate that chronic exposure to environmental Cd may promote tumorigenesis in the lung, kidney, pancreas, and breast. There is increasing evidence that Cd may contribute to the development of hepatocellular carcinoma, particularly in Asian populations. 

Non-tumorigenic human cells, exposed to relatively low levels of Cd transformed to malignant cells after the acquisition of tolerance to Cd-induced cell death. A reduction of intracellular Cd concentration mediated by the ZnT1 efflux transporter has been identified as one of the toxicity tolerance mechanisms.

Avoidance of foods containing high Cd levels is important, but so is the sufficient consumption of dietary essential metals (iron, calcium, and zinc), as well as the maintenance of adequate body iron stores and optimal body weight. Iron supplements could be a logical intervention to reduce Cd absorption and its accumulation in breast tissues, especially in highly susceptible subpopulations. 

There is no safe level of Cd exposure, and thus public health resources that promote the cessation of smoking and educate consumers about foods known to contain high levels of Cd are likely to have significant health benefits.

## Figures and Tables

**Figure 1 ijms-25-01851-f001:**
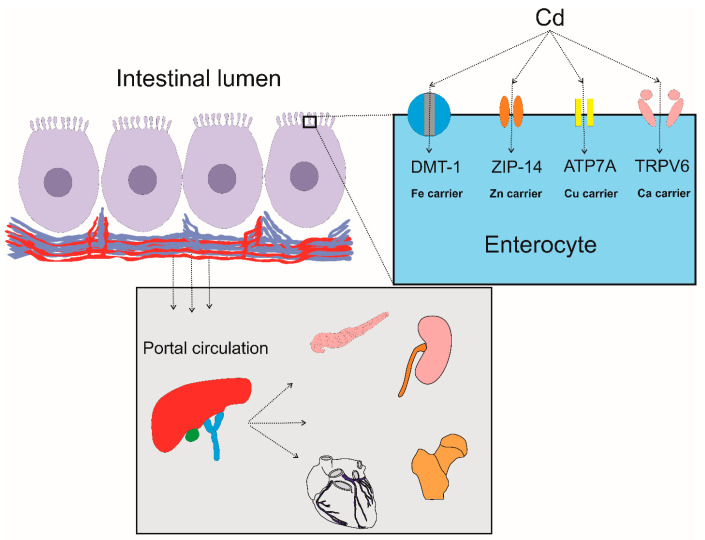
The intestinal absorption of cadmium. Cd in the diet is absorbed into the portal blood circulation, delivered to liver first, and then to cells throughout the body. The most likely metal carriers involved in Cd absorption are those for iron, zinc, copper, and calcium (Fe, Zn, Cu, and Ca) [42,43,44,45,46]. Abbreviations: DMT1, divalent metal transporter1; ZIP14, Zrt- and Irt-related protein 10; ATP7A, ATPases (Cu-ATPases) ATP7A; TRPV6, transient receptor potential vanilloid6.

**Figure 2 ijms-25-01851-f002:**
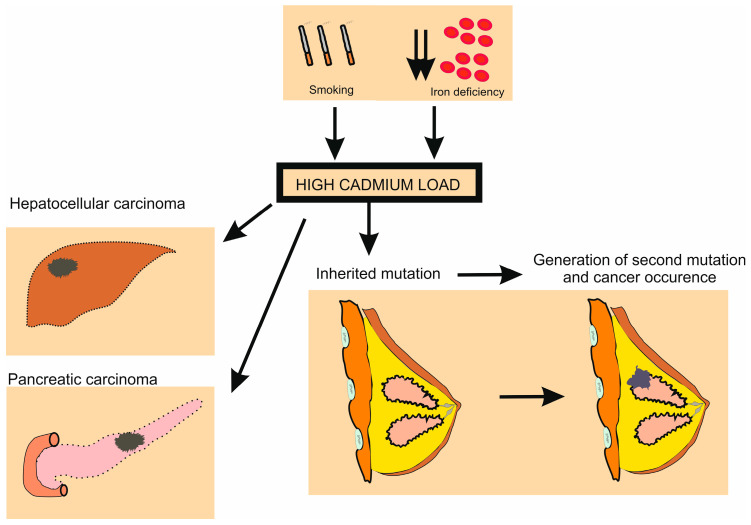
Cadmium as a multi-tissue carcinogen. Cd exposure may induce tumorigenesis in the liver, pancreas, and breast. Cd accumulation in breast tissue may contribute to the occurrence of the second mutation of the BRCA1 or BRCA2 gene with a resultant loss of heterozygosity and expression of cancer cell phenotypes.

**Table 1 ijms-25-01851-t001:** Dietary cadmium exposure estimated from food cadmium concentrations.

Foods	Cd, mg/kg		Intake Rate, g/Day	Exposure, µg/Day	
	Maximum	Typical		High	Average
Vegetables; potatoes, included	0.1	0.05	250	25	12.5
Cereals, pulses, legumes; rice, wheat grain included	0.2	0.05	200	40	10
Fruit	0.05	0.01	150	7.5	1.5
Oilseeds and cocoa beans	1.0	0.5	1	1	0.5
Meat; cattle, poultry, pig, sheep	0.1	0.02	150	15	3.0
Liver; cattle, poultry, pig, sheep	0.5	0.1	5	2.5	0.5
Kidney; cattle, poultry, pig, sheep	2.0	0.5	1	2	0.5
Fish	0.05	0.02	30	1.5	0.6
Crustaceans, mollusks	2	0.25	3	6	0.75
TOTAL	−	−	−	93.5	30

Food types and intake rates were based on a typical Australian diet [33].

**Table 2 ijms-25-01851-t002:** Environmental cadmium exposure and an increased cancer risk.

Cancer	Exposure and Risk Estimates	References
Lung	Comparing the highest vs. lowest category of urinary Cd, OR (95% CI) values were 1.68 (1.47–1.92) and 1.22 (1.13–1.31) for lung cancer and cancer mortality, respectively. The average urinary Cd excretions across three cohorts ranged from 0.25 to 0.93 µg/g creatinine.	Nawrot et al., 2015 [81]
Lung	Comparing the highest vs. lowest Cd dose category, OR (95% CI) values were 1.42 (0.91−2.23), 0.68 (0.33−1.41), and 1.61 (0.94−2.75) in the general population, occupationally exposed population, and case–control studies, respectively.	Chen et al., 2016 [82]
Kidney	Comparing the highest vs. lowest category of Cd dose metrics, the OR (95% CI) was 1.47 (1.27−1.71) for renal cancer.	Song et al., 2015 [83]
Pancreas	Comparing the highest vs. lowest category of Cd dose metrics, RR (95% CI) values were 2.05 (1.58−2.66), 1.78 (1.04–3.05), and 1.02 (0.63–1.65) for pancreatic cancer in all subjects, men, and women, respectively.	Chen et al., 2015 [84]
Breast	Comparing the highest vs. lowest category of urinary Cd, the OR (95% CI) was 2.24 (1.50–3.34) for breast cancer. For each 0.5-µg/g creatinine increase of urinary Cd, the OR for breast cancer was increased 1.66-fold (95% CI: 1.23–2.25).	Larsson et al., 2015 [85]
Breast	Comparing the highest vs. lowest quartile of urinary Cd, the OR (95% CI) was 2.24 (1.49–3.35) for breast cancer. An association of dietary Cd exposure and breast cancer was insignificant (RR 1.01, 95% CI: 0.89–1.15).	Lin et al., 2016 [86]
Breast	Mean (95% CI) values for Cd and Ni were 2.65 (1.57–3.73) and 2.06 (1.20–3.32) higher in scalp hair and toenail samples from women with breast cancer, compared to controls.	Jouybari et al., 2018 [87]
Breast	Among postmenopausal women, dietary Cd exposure level was not associated with breast cancer risk (RR, 1.03, 95% CI: 0.89–1.19)	Van Maele-Fabry et al., 2016 [88]

OR, odds ratio; CI, confidence interval; RR, relative risk; Ni = nickel.

**Table 3 ijms-25-01851-t003:** Levels of environmental cadmium exposure and effects on the liver.

Study Population	Exposure Levels and Effects Observed	Reference
United States NHANES 1988–1994 n 12,732, ≥20 years	Urinary Cd levels ≥ 0.83 μg/g creatinine were associated with a 1.26-fold increase in risk of liver inflammation in women. Urinary Cd levels ≥ 0.65 μg/g creatinine were associated with liver inflammation (OR 2.21), NAFLD (OR 1.30), and NASH (OR 1.95) in men	Hyder et al., 2013 [109]
United States NHANES 1999–2015 n 11,838, ≥20 years	A ten-fold increment of urinary Cd was associated with elevated plasma levels of ALT (OR 1.36) and AST (OR 1.31).	Hong et al., 2021 [110]
United States NHANES 1999–2016 n 4411 adolescents	Urinary Cd quartile 4 was associated with elevated plasma ALT (OR 1.40) and AST (OR 1.64). The liver effect of Cd was larger in boys than girls.	Xu et al., 2022 [111]
South Korea, n 3914, ≥19 years	Blood Cd levels ≥ 1.98 μg/L were associated with elevated plasma levels of AST (OR 3.61) and ALT (OR 2.40).	Kang et al., 2013 [112]
Korean National Environmental Health Survey (2015–2017) n 2953, ≥19 years	Geometric mean urinary Cd was 0.42 µg/L. The top urinary Cd quartile (>0.88 μg/L) was associated with higher plasma AST and GGT, compared with quartile 1. Urinary Cd quartiles 4, 3, and 2 were associated with higher plasma ALT, compared with the bottom quartile.	Kim et al., 2021 [113]
South Korea, n 12,099, ≥19 years	Blood Cd levels higher than 0.96 and 1.41 μg/L were associated with NAFLD (OR 1.91) and NASH (OR 1.41).	Park et al., 2021 [114]
Korean NHANES 2008–2017 n 15,783, ≥20 years, mean age 46	Serum Cd levels >1.41 µg/dL were respectively associated with 1.90-, 1.26-, 1.73- and 2.53-fold increases in risk of ALT elevation, hepatic steatosis, fibrosis, and AST elevation, compared to serum Cd levels < 0.651 µg/dL	Han et al., 2022 [115]
South Korea Baseline (2011), n 2086 adults Follow-up (2015), n 503 adults	The geometric mean blood Cd was higher in females than males (1.16 vs. 0.96 μg/L). No change in blood Cd levels in both genders over a 5-year period. ALT and GGT rose only in females.	Seo et al., 2023 [116]

NHANES, National Health and Nutrition Examination Survey; n, sample size; OR, odds ratio; ALT, alanine aminotransferase; AST, aspartate aminotransferase; GGT, γ-glutamyl transferase; NAFLD, non-alcoholic fatty liver disease; NASH, non-alcoholic steatohepatitis.

**Table 4 ijms-25-01851-t004:** The genesis of cancer cells after prolonged exposure to cadmium.

Immortal Cell Line, Reference	[Cd], Exposure Duration	Transformant and Tumor Characteristics
Human urothelium, UROtsa [146,147]	1 μM Cd^2+^ 8–10 weeks.	Soft agar colony formation. Tumorigenic in nude mice with the transitional cell carcinoma phenotypes; gene expression profiles like the basal subtype of muscle invasive bladder carcinomas. Acquired Cd tolerance as an adaptive survival mechanism.
Human breast epithelium, MCF-10A [148]	2.5 μM Cd^2+^ 40 weeks.	Basal-like breast cancer cells; ER-α-negative, HER2-negative, diminished BRCA1 expression, persistent cell proliferation, elevated expression of cytokeratin 5 and p63, overexpressed c-myc and Kras, and displayed a global DNA hypomethylation state. Tumors were ER-α-negative.
Human bronchial epithelium, BEAS-2B [149]	1 µM Cd^2+^ 6 months. Passage-matched cells as controls.	Activated ERK and AKT signaling, elevated HIF-1 and VEGF expression. Inhibition of ROS generation attenuated the activation of ERK, AKT, p70S6K1, and the expression of HIF-1α. Tumorigenic in nude mice. Formation of tubes in an in vitro test for angiogenesis.
Human peripheral lung epithelium, HPL-1D [150]	5 μM Cd^2+^ 20 weeks.	Soft agar colony formation. Decreased expression of the tumor suppressor genes p16 and SLC38A3, increased expression of the oncoproteins KRAS and NRAS and vimentin, the epithelial-to-mesenchymal transition marker, overexpressed MT-1A, MT-2A, HO-1, HIF-1A, ZnT1, ZnT5, and ZIP8. Acquired Cd tolerance as an adaptive survival mechanism.
Human pancreatic ductal epithelium, HPDE [151]	1 μM Cd^2+^ 29 weeks.	Increased secretion of matrix metalloproteinase-9 (MMP-9) and over-expressed the pancreatic cancer marker S100P. Formation of poorly differentiated glandular-like structures on soft agar.
Rat liver epithelium, TRL1215 [152,153].	2.5 μM Cd^2+^ for 10 weeks. 1 μM Cd^2+^ for 28 weeks.	Hyperproliferation, increased invasiveness, reduced dependency on serum, increased DNA methylation, enhanced DNA methyltransferase activity. Over expression of c-myc and c-jun oncogenes.

[Cd], cadmium concentration; ER-α, estrogen receptor-alpha; HER2, human epidermal growth factor receptor 2; BRCA1, breast cancer susceptibility gene 1; BERK, extracellular-signal-regulated kinases; HIF-1, hypoxia-inducible factor-1; MMP-9, matrix metalloproteinase-9; MT, metallothionein; HO-1, heme oxygenase-1.

## Data Availability

Not applicable.

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
