# Peer review of "Toxicity Tolerance in the Carcinogenesis of Environmental Cadmium"

_ijms, 2024, doi:10.3390/ijms25031851_

Round 1

Reviewer 1 Report

Comments and Suggestions for Authors

1:Lines 59-60, active and passive smoking should be one of the main causes of cadmium's health effects.

2:Line 88, 2.2 The title is the intestinal absorption rate of cadmium, but the content does not match the content of the title, you can change the title, or increase the content related to the intestinal cadmium absorption rate.

3:Lines 117 and 136 are incorrectly numbered.

4:Figure 2 can be supplemented with cadmium-induced cancers of other organs.

5: Lines 301-303 should be annotated at the beginning of the list

6:Figure 2 does not show that cadmium accumulation leads to the occurrence of BRCA1 or BRCA2 gene mutations, and should cadmium be reflected in Figure.

7:Part of line 471 could be adjusted and should be placed before 4.1.

8:Line 503, 4.4 headings are broad, but the content is mainly zinc transporters, and it is recommended to change the subheadings.

Author Response

Reviewer 1

We thank Reviewer for evaluating our work, knowledgeable comments, and suggestions for improvement. We have addressed point-by-point issues and concerned raised as detailed below. Changes to the text in a manuscript are in blue

Comments and Suggestions

1: Lines 59-60, active and passive smoking should be one of the main causes of cadmium's health effects.

RESPONSE:

  • “Active and passive smoking” as sources of Cd have been inserted in second sentence of abstract and in the manuscript (lines32-33).

2: Line 88, 2.2 The title is the intestinal absorption rate of cadmium, but the content does not match the content of the title, you can change the title, or increase the content related to the intestinal cadmium absorption rate.

RESPONSE:

  • “Rate” has been deleted and a subtitle now is, 2.2. The intestinal absorption of cadmium.

3: Lines 117 and 136 are incorrectly numbered.

RESPONSE:

  • Errors have been corrected.
  • Line 117 now reads, 2.3. Metal Transporters Involved in Cadmium Absorption.
  • Line 136 has been changed to 2.4. Determinants of Cadmium Absorption

all subtitles of 2.4 have accordingly been changed as below.

2.4. Determinants of Cadmium Absorption

2.4.1. Body Iron Stores and Iron Deficiency 

2.4.2. Dietary Factors: Diet Quality

2.4.3. Genetic Factors: The ZIP8, ZIP14, TfR and H63D variants

2.5. Summary: Determinants of Cadmium Absorption

4: Figure 2 can be supplemented with cadmium-induced cancers of other organs.

RESPONSE:

  • Liver and pancreas have been added to Figure 2.

5: Lines 301-303 should be annotated at the beginning of the list.

RESPONSE:

  • On line 56, a full spelling of NHANES has been given in its first appearance.
  • In effect, we are required to maintain the table footnote according to the mandate of JMSC (journal style).

6: Figure 2 does not show that cadmium accumulation leads to the occurrence of BRCA1 or BRCA2 gene mutations, and should cadmium be reflected in Figure.

RESPONSE:

  • We have made changes to Figure 2 to convey the following ideas.
  • Cadmium may induce tumorigenesis in liver, pancreas, and breast.
  • “HIGH CADMIUM LOAD” is a major contributing factor in the occurrence second mutation of BRCA1 or BRCA2 gene.
  • Legend to Figure 2 has been changed to below.

Figure 2. Cadmium as a multi-tissue carcinogen. Cd exposure may induce tumorigenesis in liver, pancreas, and breast. Cd accumulation in breast tissue may contribute to the occurrence of second mutation of the BRCA1 or BRCA2 gene with resultant loss of heterozygosity and expression of cancer cell phenotypes.

7: Part of line 471 could be adjusted and should be placed before 4.1.

RESPONSE:  Two statements quote below have been inserted in an introduction part above Section 4.1 (lines 442-444).

  • However, a high cost involved in a conduct of 2-year bioassay has led to an idea of generating cancer cells from non-tumorigenic human cells. Accordingly, this section summarizes a such successful attempt to produce cancer cells under in vitro conditions.

8: Line 503, 4.4 headings are broad, but the content is mainly zinc transporters, and it is recommended to change the subheadings.

RESPONSE:

  • The refereed subtitle has now been changed to 4.4. Acquired Resistance to Cadmium-Induced Cell Death: Role of ZnT1.

Reviewer 2 Report

Comments and Suggestions for Authors

Title: Toxicity Tolerance in the Carcinogenesis of Environmental Cadmium

On account of the manuscript ijms-2847415, entitled “Toxicity Tolerance in the Carcinogenesis of Environmental Cadmium” by Aleksandar Cirovic and Soisungwan Satarug, the authors reviewed the dietary sources and determinants on the body burden of Cd, the epidemiological and experimental studies of Cd as a multi-tissue carcinogen. The author highlights epidemiological studies that connected an enhanced risk of various neoplastic diseases to chronic exposure to environmental Cd, especially for the impact of body iron stores on the absorption rate of Cd. The manuscript was well written and designed, and the authors provide valuable information for this research field. Great work and much effort were given. Nevertheless, before this manuscript can be recommended to publication, it needs to be improved. Please find my comments and suggestions below.  

1. In abstract: line 11-21, it should be succinct with less words.

2. Some necessary explanation about meta-analysis is needed, such as OR and RR in Table 2.

3. In line 273-275, it seems to have little relationship with the theme Cadmium and Hepatocellular Carcinoma.

4. In line 354-357, are you sure that this is in Korean study? It seems to have weak causal relationship between line 355 and line 356.

5. In conclusions: line 538-564, please don’t simply repeat something that were in your paper. It should be a brief summary of the manuscript’s main points.

Comments on the Quality of English Language

There is no comments about this.

Author Response

Reviewer 2

We thank Reviewer for evaluating our work, insightful comments, guidance, and suggestions for improvement. Point-by-point responses to issues and concerns raised are detailed below. Changes to the text in a manuscript are in blue

Comments and Suggestions

Title: Toxicity Tolerance in the Carcinogenesis of Environmental Cadmium

On account of the manuscript ijms-2847415, entitled “Toxicity Tolerance in the Carcinogenesis of Environmental Cadmium” by Aleksandar Cirovic and Soisungwan Satarug, the authors reviewed the dietary sources and determinants on the body burden of Cd, the epidemiological and experimental studies of Cd as a multi-tissue carcinogen. The author highlights epidemiological studies that connected an enhanced risk of various neoplastic diseases to chronic exposure to environmental Cd, especially for the impact of body iron stores on the absorption rate of Cd. The manuscript was well written and designed, and the authors provide valuable information for this research field. Great work and much effort were given. Nevertheless, before this manuscript can be recommended to publication, it needs to be improved. Please find my comments and suggestions below. 

  1. In abstract: line 11-21, it should be succinct with less words.

RESPONSE:

  • Background information in abstract has been described in brief, and total word count of abstract now is 226.

  1. Some necessary explanation about meta-analysis is needed, such as OR and RR in Table 2.

RESPONSE:

  • We made changes to the presentation of results from meta-analysis to make them self-explanation with respect to OR (95%CI) and RR (95% CI) values.
  • The main purpose of this Table is to show Cd as a multi-tissue carcinogen rather than comparing an effect size of Cd across different types of cancer. Therefore, we made no attempt to explain OR/RR values for each cancer.

  1. In line 273-275, it seems to have little relationship with the theme “Cadmium and Hepatocellular Carcinoma”.

RESPONSE:

  • We have inserted non-alcoholic steatohepatitis (NASH) as another known cause of HCC. We also added studies from U.S. and Korean consistently that provide evidence linking HCC to Cd exposure. The referred paragraph now reads as below.

Other HCC causative conditions are alcoholic steatohepatitis, non-alcoholic steatohepatitis (NASH), iron overload, excessive copper accumulation (Wilson-disease), congenital α1-antitripsin deficiency, autoimmune hepatitis, acquired and congenital glycogen storage disease [96-98]. Studies from United States and Korea, described below, observed associations of increased risk of NASH and Cd exposure, thereby suggesting that Cd may promote NASH development, which may progress to liver fibrosis and HCC.

  1. In line 354-357, are you sure that this is in Korean study? It seems to have weak causal relationship between line 355 and line 356.

RESPONSE:

  • It is confirmed that both studies were conducted in South Korea.
  • We have added below sentence to explain an increased risk of breast cancer among women with anemia who lived in urban area, compared to residents of rural communities (lines 364-366).
  • Collectively, data from these two Korean studies could be interpreted to suggest that an increased risk of breast cancer among anemic women who lived in urban communities was due an increased exposure to Cd in dust.

  1. In conclusions: line 538-564, please don’t simply repeat something that were in your paper. It should be a brief summary of the manuscript’s main points.

RESPONSE:

  • Conclusion has been rewritten to reflect main findings as quoted below. No duplication of results that have already provided. Public health implications have been maintained.

Epidemiologic and experimental data provide compelling evidence that Cd is a multi-tissue carcinogen. Systematic review and meta-analyses indicate that chronic exposure to environmental Cd may promote tumorigenesis in lung, kidney, pancreas, and breast. There is increasing evidence that Cd may contribute to the development of hepatocellular carcinoma, particularly in Asian populations.

Non-tumorigenic human cells, exposed to relatively low levels of Cd transformed to malignant cells after acquisition of tolerance to Cd-induced cell death. A reduction of intracellular Cd concentration mediated by ZnT1 efflux transporter has been identified as a one of the toxicity tolerance mechanisms.

Reviewer 3 Report

Comments and Suggestions for Authors

Dear authors

Thank you very much for the interesting review about cadmium effects on organisms. here are my questions, which , I think , should improve your interesting manuscript.

Q1: In the text you only mentioned  Australian diet as an example of dietary exposure. Please can you add some other example of diets, e.g. European, USA, Mediterranean...etc to compare possible daily income?

Q2: Please can you add more info about therapy of Cd exposure--chelation?

Q3: Do you think that the lack of iron compensates the body with an increased intake of cadmium?

Q4: Is the amount of Cd in sea products determined and tested? Beacuse it looks like they are one of the main sources of cadmium.

Q5: Is there a set amount of cadmium in food at all, because you wrote that there is no safe level of exposure to Cd.

Q6: What happens to food which has a high Cd content? How is it neutralized?

Author Response

Reviewer 3.

We thank Reviewer for evaluating our work, knowledgeable comments, and suggestions for improvement. Responses to reviewer questions are provided below. Changes to the text in a manuscript are in blue

Comments and Suggestions

Thank you very much for the interesting review about cadmium effects on organisms. Here are my questions, which, I think, should improve your interesting manuscript.

Q1: In the text you only mentioned Australian diet as an example of dietary exposure. Please can you add some other examples of diets, e.g. European, USA, Mediterranean...etc to compare possible daily intake?

RESPONSE:

  • We have provided dietary exposure in Sweden and France that were estimated by total diet study, same as Australia study (lines 86-91 ) as quoted below.

Average dietary Cd exposure level in Sweden was 10.6 μg/day with 40-50% of Cd coming from staple foods (potatoes and wheat), while a high dietary Cd exposure level was 23 μg/day, with additional Cd coming from seafood and spinach [3]. Average dietary Cd exposure in France was 11.2 µg/day with 35% of Cd coming from bread products and another 26% from potatoes-based products, while a high Cd exposure level was 18.9 µg/day with additional Cd coming from mollusks and crustaceans [4].

Q2: Please can you add more info about therapy of Cd exposure--chelation?

RESPONSE:

  • As we have indicated (lines 138-139), there is no effective chelation therapy for Cd, and the most practical preventive measure is to minimize Cd absorption rate and avoidance of high-Cd foods.

Q3: Do you think that the lack of iron compensates the body with an increased intake of cadmium?

RESPONSE:

  • This question is addressed in 2.5. Summary: Determinants of Cadmium Absorption, where it is stated that body iron store status is a major determinant of Cd absorption rate, and thus Cd body burden. When iron status of the body is low, the intestinal absorption rate rises, a compensatory mechanism to acquire more iron. A systematic review [ref 79] has confirm that both zinc and iron status are equally important [79]. This result can be expected as it is now known that absorption of iron and zinc is mediated by the same metal transporters (detailed in Section 2.3. Metal Transporters Involved in Cadmium Absorption).

Q4: Is the amount of Cd in sea products determined and tested? Because it looks like they are one of the main sources of cadmium.

RESPONSE:

  • Reviewer has correctly identified major Cd sources which have been now added, lines 86-9 Please see response to Q1.
  • As part of Total Diet Study, Food Authority Agency, collected food samples, included seafoods and analyzed for various contaminants and food additives. This is a food safety monitoring program.

Q5: Is there a set amount of cadmium in food at all, because you wrote that there is no safe level of exposure to Cd.

RESPONSE:

  • There are food regulatory bodies that are responsible for setting permissible levels of Cd in marketable products. For example, European Food Safety Agency has set maximally permissible levels of Cd and Pb in various food items.

Q6: What happens to food which has a high Cd content? How is it neutralized?

RESPONSE:

  • High Cd levels in food is a substantial problem with no identified solution. There have been continuing efforts to produce new variety of rice/wheat/potato (staple foods) that least absorb Cd from soils. Changes in agricultural practice for growing rice crop like flooding of paddy soils has been shown to have some promise. Nevertheless, to reduce intestinal absorption of Cd from foods is a viable option as is avoidance of foods containing high Cd.
  • There were some speculations regarding food Ca (calcium) biofortification. This food fortification faces limitations (Li HB, Xue RY, Lin XY, Ma LQ. Responses to Comments on "Cadmium oral bioavailability is affected by calcium and phytate contents in food: Evidence from leafy vegetables in mice". J Hazard Mater. 2022 Sep 15;438:129497. doi: 10.1016/j.jhazmat.2022.129497).
  • This question is addressed in part in section 2.4.2. Dietary Factors: Diet Quality (lines 168-213).
